# Functions of Circular RNA in Human Diseases and Illnesses

**DOI:** 10.3390/ncrna9040038

**Published:** 2023-07-04

**Authors:** Alison Gu, Dabbu Kumar Jaijyan, Shaomin Yang, Mulan Zeng, Shaokai Pei, Hua Zhu

**Affiliations:** 1Department of Microbiology and Molecular Genetics, New Jersey Medical School, Rutgers University, 225 Warren Street, Newark, NJ 070101, USAsp2326@gsbs.rutgers.edu (S.P.); 2Department of Pain Medicine and Shenzhen Municipal Key Laboratory for Pain Medicine, Huazhong University of Science and Technology Union Shenzhen Hospital, Shenzhen 518052, China

**Keywords:** circular RNA, cancer, disease diagnosis, long non-coding RNAs (lncRNAs), micro-RNA, miRNA sponges, protein decoy, viral circular RNA

## Abstract

Circular RNAs (circRNAs) represent single-stranded RNA species that contain covalently closed 3′ and 5′ ends that provide them more stability than linear RNA, which has free ends. Emerging evidence indicates that circRNAs perform essential functions in many DNA viruses, including coronaviruses, Epstein–Barr viruses, cytomegalovirus, and Kaposi sarcoma viruses. Recent studies have confirmed that circRNAs are present in viruses, including DNA and RNA viruses, and play various important functions such as evading host immune response, disease pathogenesis, protein translation, miRNA sponges, regulating cell proliferation, and virus replication. Studies have confirmed that circRNAs can be biological signatures or pathological markers for autoimmune diseases, neurological diseases, and cancers. However, our understanding of circRNAs in DNA and RNA viruses is still limited, and functional evaluation of viral and host circRNAs is essential to completely understand their biological functions. In the present review, we describe the metabolism and cellular roles of circRNA, including its roles in various diseases and viral and cellular circRNA functions. Circular RNAs are found to interact with RNA, proteins, and DNA, and thus can modulate cellular processes, including translation, transcription, splicing, and other functions. Circular RNAs interfere with various signaling pathways and take part in vital functions in various biological, physiological, cellular, and pathophysiological processes. We also summarize recent evidence demonstrating cellular and viral circRNA’s roles in DNA and RNA viruses in this growing field of research.

## 1. Introduction

Circular RNA creates a continuous loop that is covalently closed and is a form of single-stranded RNA [1]. Backsplicing produces circRNAs when a downstream splice-donor site covalently links together with an upstream splice-acceptor site [2]. The first circRNA molecules that were discovered were viroids even though they were not produced through backsplicing [3]. CircRNAs were then analyzed in the cytoplasm of cells by high-resolution electron microscopy, but they were mostly thought to be debris created by abnormal splicing events [4,5]. Recently, circRNAs have been found in rats, humans, fungi, mice, and other organisms through transcriptomic sequencing (RNA-seq) and circRNA-specific bioinformatics tools [6]. CircRNAs are found to have expressed in specific tissue and display tissue specificity [7]. Even though many circRNAs are generated from pre-mRNA molecules during splicing, they are often grouped in long non-coding RNAs (lncRNAs). CircRNAs also act as protein or RNA decays that control the expression of the gene, which is seen analogously in different lncRNAs. One of the most important functions of circRNAs is their interaction with miRNAs. Single circRNAs can have numerous binding sites for miRNA and serve as a “sponge” to suppress the biological functions of miRNAs [8]. A recent study showed that pancreatic cancer tissues consist of upregulated circZFR, causing increased tumorigenicity and promoting cell proliferation. A study showed that circZFR sponging miR-375 caused the JNK signaling pathway to initiate GREM2 upregulation in cells found in the pancreatic cancer [9]. Studies have found that small nucleolar RNA host gene 1 (SNHG1) is a lncRNA that is regulated in the development of bladder cancer. Expression of EZH2 in the nucleus is increased by SNHG1, causing CDH1 to downregulate. This causes levels of E-cadherin to reduce and aid metastasis of cells in the bladder cancer [10].

CircRNAs have a unique structure that gives them more stability, a better half-life, and increased RNase R resistance as compared to linear RNAs (Figure 1) [11]. Many circRNAs are generated in the cells from genes that code for proteins and contain single or several exons (Figure 1) [12]. Leftover debris generated from alternative splicing of linear RNA can be found within circRNA, but some circRNAs consist of exons that are not present in linear RNA transcripts [7]. Even though circRNAs lack 5′-3′ ends, known as polyadenylated (polyA) tails, they are mostly localized to the cytoplasm [13]. However, exon-intron circRNAs are produced when internal introns are retained [14]. If intronic lariats do not debranch during canonical splicing, they can produce intronic circular RNAs (ciRNAs) [15]. Additionally, circular RNA can regulate the transcription of parental genes and are present in the nucleus [14].

Functions of circRNA include but are not limited to miRNA inhibition, epithelial-mesenchymal transition, and tumorigenesis [16,17,18]. Expression of circRNA can also be specific to the tissue [17]. Furthermore, circRNAs were found in different types of biological samples including plasma, cell-free saliva, and exosomes [19]. CircRNAs have been found to play important roles in age-associated diseases, including cancer [20], diabetes [21], neurodegenerative diseases [22], osteoarthritis [23], age-related diseases [24], cardiovascular diseases [25], stress [26], and more (Figure 2). However, circRNA’s roles in viruses are well not described. As a result, circRNAs may be important pathological biomarkers or drug targets for various diseases, including autoimmune diseases, cancers, neurological diseases, and more (Figure 3). Circular RNAs also have various biomedical applications (Figure 3). In various DNA viruses such as Kaposi sarcoma virus (KSHV), Epstein–Barr virus, and human papillomaviruses (HPVs), viral-encoded circRNAs were found to be present. Further findings demonstrated that circRNA is a critical part of the coronavirus transcriptome [1]. Despite there being studies about circRNAs being potential biomarkers and therapeutic targets, the mechanisms of how circRNAs implement physiological and pathological roles in disease remain insufficiently understood.

Several studies have been published in recent years about circRNA’s roles in DNA and RNA viruses. Considering the growing field of circRNA research, this review will be examining the function of circRNAs in DNA and RNA viruses. However, to fully understand the function of circRNAs in these viruses, we must first discuss how circRNAs metabolize, their functions, and how circRNAs interact with proteins first.

### Circular RNA (circRNA) and Long Non-Coding RNA (lncRNA)

circRNA and lncRNA represent two different types of noncoding RNA species that usually do not code for any proteins [27]. CircRNAs are covalently closed RNA molecules without any 3′ tail or 5′ cap. circRNAs can have profound effects on gene transcription and mRNA stability because they interact with RNA-binding proteins. Interestingly, some circRNAs have been found to encode proteins [28,29]. Similar to lncRNA, circRNAs also contain miRNA response elements (MRE) and function as a miRNA sponge [30]. circRNAs also function as competing endogenous RNA and suppress the interaction between miRNA and their target genes. Many studies have confirmed that circRNAs can regulate pyroptosis, autophagy, cell proliferation, and differentiation [31].

A major difference between lncRNA and circRNA is their structure and origin. LncRNAs are spliced, capped, and polyadenylated with no ability to code protein. The length of lncRNAs is more than 200 nt and is considered to be part of a group of ncRNAs [32]. LncRNAs were created from different genome regions, such as enhancers; intergenic regions, strands that are the opposite of protein-coding genes; and promoter regions [33]. Furthermore, lncRNAs arise through various processes, including (1) tandem duplication, (2) retrotransposition, (3) chromosomes rearrangement, (4) insertion of sequences that are transposable, and (5) metamorphosis from sequences coded with protein that already exists [34]. lncRNAs have been found to regulate RNA metabolism, translation, transcription, apoptosis, and other cellular functions. lLncRNAs can be categorized into antisense lncRNA, intronic lncRNA, sense lncRNA, intergenic lncRNA, and bidirectional lncRNA [35]. LncRNAs bind with the 3′ UTRs of mRNA molecules to regulate translation and gene transcription [36]. lLncRNAs contain miRNA response elements (MRE) and interact with miRNAs. Moreover, lncRNAs are found to act as competing endogenous RNAs and function as miRNA sponges, therefore reducing miRNA activity [37,38]. lncRNAs can impair the association between miRNA and their gene targets.

## 2. Metabolism of Circular RNA

### Biogenesis, Localization, Modification, and Regulation of CircRNAs

RNA polymerase II synthesizes pre-mRNA that consists of exons and introns. This pre-mRNA molecule has a 7-methylguanosine at the 5′ end and a poly-adenosine tail at the 3′ end. The genes are interrupted by the presence of introns in the eukaryotic cells; therefore, the pre-mRNA molecules must undergo splicing to remove these noninformational intronic sequences. Additionally, pre-mRNA produces a translatable mature mRNA molecule through splicing at canonical splicing sites and spliceosomes. As previously mentioned, the biogenesis of circRNA has been found to be dependent on spliceosomes, canonical splicing machinery, and splice signal sites [39]. RNA binding proteins (RBPs) (trans-factor), and intronic complementary sequences (ICSs) (cis factor) have essential functions in the formation of circRNAs by bringing upstream acceptor sites close to the downstream donor site [40]. It has been found that hindering pre-mRNA processing in cells causes the production of genes to circRNAs, suggesting that linear RNA and respective circRNAs compete with each other [41].

CircRNAs are produced by backsplicing, meaning that the 3′ and 5′ ends of an exon are ligated to create a sealed circular molecule that consists of a site of backsplicing junction region. It also means that the circular RNA structure is formed through an exon that has upstream 3′, 5′-phosphodiester bonds [42]. This shows that circRNA formation is contingent on the cellular canonical splicing machinery [40]. Originally, circRNAs were thought of as splicing errors consisting of “scrambled exons” [43]. However, it was found that suppressing the spliceosome (large RNA-protein complex) via draining the units of the U2 snRNP raises the ratio of circular RNAs to linear RNAs in *Drosophila melanogaster* cells. This demonstrates that growing RNA can be processed via other alternative cellular pathways that expedite backsplicing events when the processing of pre-mRNA is reduced [41]. Studies have shown that the biogenesis of circRNAs is regulated by RBPs that bind with flanking intron sequences. The RBP binding to the intron sequences brings introns adjacent to each other that promote circRNA formation. It was found that RBP binding to the introns can maintain or damage Alu pairs to promote or inhibit backsplicing accordingly [17,44,45].

Studies have shown that mammalian-wide interspersed repeat (MIR)-mediated RNA circularization is required to create mammalian circRNAs [46]. Moreover, excellent studies have also found that exon circularization is also promoted by Alu repeats in introns flanking the exon sequence. The pairs in the identical intron sequence facilitate the expulsion and canonical splicing [47]. A study described that circularization requires circRNAs formation to be regulated by a functional processing signal in the 3′ end that is required for circularization. Moreover, exon sequences and repeats in introns promote circularization. The circularization process occurs post-transcriptionally [48].

CircRNAs are abundant and stable notably in various cell types [49]. Recent studies have provided insight into how circRNA degrades. One study demonstrates that endogenous cricRNAs are more likely to produce imperfect intra-dsRNAs and favor binding to PKR, thus making it constitute an inhibitor of endogenous PKR. Another study found that HRSP12, RNase P/MRP, and YTHDF2 cleave m6A-containing circRNAs. Accelerated degradation of YTDHDF2-bound RNAs occurs when YTHDF2 and RNase P/MRP have HRSP12 acting as a bridge between them [50]. Interestingly, recent research has also found that circRNA abundance decreased in colon cancer and other highly proliferative cancers compared to normal tissue. This means that there is a negative correlation between proliferation and circRNA abundance [51]. During the initial stages of the innate immune response, circRNAs sustain RNase L degradation, which is associated with SLE [52].

A recent study found that one way cells could clear circRNAs is to secrete them using extracellular vesicles (EVs). CircRNAs are more enhanced compared to linear forms of identical genes and readily identified in EV preparation, thus supporting the idea that clearing circRNAs via EVs is a mechanism that cells can use [53]. Studies have shown that GW182 expedites the accumulation of P-body and functions as a molecular scaffold, thus making it a critical part of P-bodies and RNAi machines. The abundance of mature circRNAs stayed the same when core structures of P-bodies and RNAi machines were reduced, which means that circRNA degradation may not need P-bodies and RNAi machines [54]. Due to their stability, circRNAs can aggregate; however, processes that clear circRNA from cells are obscure.

Several researchers have found that circRNAs increase in number during murine spermatogenesis. Backsplicing in a subgroup of circRNAs takes place in areas that are enhanced with N6-methyladenoise (m^6^A). Concurrently, YTHDF3 identifies an m^6^A-modified start codon and interferes with the translation initiation [55]. Further research found that some m^6^As control the buildup of circ-ZNF609. Circ-ZNF609 backsplicing reaction must be directed by YTHDC1 and METTL3. There were substantial correlations found amid the capability of m^6^A exons tolerating backsplicing, needing METTL3, and the binding of YTHDC1 [56].

Exonic circRNAs are commonly localized in the cytoplasm [13]. Additionally, recent studies have identified that circRNAs containing introns play a role in controlling their parental gene and being maintained in the nucleus. One study detected a group of circRNAs known as exon-intron circRNAs (ElciRNAs) that work with U1 snRNP to increase parental gene expression [14]. Another study found that the knockdown of circular intronic circRNAs (ciRNAs), also mostly found in the nucleus, caused the parent gene expression to decrease [15]. Recent research has also found that circRNA localization is regulated by UAP56/URH49. Additionally, these homologs can also measure how long mature circRNAs are, thus controlling the productivity of the nuclear export [57]. In the cytoplasm, the stability of *HMGA2* mRNA was improved when the circNSUN2/IGF2BP2/*HMGA2* RNA-protein ternary complex was created by circNSUN2, thus aiding in the advancement of colorectal cancer metastasis [58]. Further research demonstrated that circNSUN2 is transported from the nucleus to the cytoplasm by YTHDC1 when it goes through m^6^A modification.

CircRNAs created from exons, known as exonic circRNAs, are rarely distributed to the nucleus. The few exonic circRNAs that are located in the nucleus enhance the nucleus’ ability to contain proteins and bring proteins to the chromatin [59,60]. In addition, circRNAs can be identified in both urine and circulation. These results demonstrated that the correlation of proliferation and circRNA abundance is negative. It was also found to be transported by extracellular vesicles (EVs). Producer cells linked with miR levels regulate exosomal circRNA sorting, whereas certain biological activities moved to recipient cells are mostly obscure in different environments [61].

## 3. Functions of Circular RNA

CircRNAs play important functions in many cellular processes, including controlling gene expression through forming complexes by binding to RNA-linked proteins and constituting miRNA sponges. Many are interested in circRNA’s roles in human diseases and cancers because of their expression in pathological conditions and tissue-specific approach [62]. Acknowledging the various roles of circRNAs as a whole is critical before examining their role in DNA and RNA viruses.

### 3.1. Cell Proliferation and CircRNAs

Controlling the cell cycle meticulously and precisely is critical during typical cellular responses to environmental cues. More and more circRNAs have been discovered to manage cellular proliferation during cell cycle checkpoint regulators, transcription factors, and signaling pathways. Studies have shown that circRNAs are involved in controlling WNT/β-catenin pathways to increase cellular proliferation. One study found that a decrease in *circHIPK3* caused FZD4 receptor and WNT2 ligand expression to diminish. This led to less cell proliferation of hampered retinal endothelial and nuclear β-catenin [63]. Additionally, circRNAs affect both AKT/PI3K and ERK/MAPK. The two most common pathways that monitor the increase of cells are AKT PI3K and ERK/MAPK [8]. Studies have shown that growth factors such as FGF attach to tyrosine kinase’s receptor in MAPK/ERK pathways. The binding causes MAPK to phosphorylate, leading to cell proliferation and ERK activation. PI3K phosphorylates AKT and stimulates cell proliferation when receptor tyrosine kinases ligands attach to each other in AKT/PI3K signaling pathways. In glioblastoma and HCC, *CDR1as* and *circNT5E* promoted cell proliferation by increasing PI3K expression [64,65]. In esophageal squamous cell carcinoma and colorectal cancer, EGFR receptor expression was enriched by *circHIPK2* and *CDR1as* [61,66]. The expression of FGF2 ligand has improved by *circWDR77* in smooth muscle cells found in vascular tissue [67]. Furthermore, a recent study found that the knockdown of hsa_circ_0064559 expanded apoptosis and reduced proliferation rate in cells found in colorectal cancer [68]. Collectively, this evidence shows that circRNAs have the ability to regulate cell proliferation by regulating various signaling pathways and thus can control important cellular processes.

### 3.2. CircRNAs Functioning as miRNA Sponges

miRNAs control post-transcriptional repression by combining with protein-coding mRNAs, indicating its crucial role in gene-regulatory roles [69]. Evidence demonstrated that miRNA sponges’ activity is unique to each individual miRNA seed family and just as successful as current antisense technology. Additionally, assay miRNA loss-of-function phenotype and predictions of different targets can be confirmed by using these miRNA sponges [70]. CircRNAs functioning as miRNA sponges have been demonstrated through a multitude of recent research [42]. Table 1 lists the circRNAs that act as microRNA sponges. Studies have found that a transcribed CDR1as/CiRS-7 from the CDR1 antisense strand is extensively found in the brains of humans or mice. While CiRS-7 absorbs miR-7, CiRS-7 boosts target miR-7 expression and blocks its biological functions at the same time. However, circRNA is split by miRNA during miRNA sponging [71]. Additionally, it has been shown that ample amounts of circRNA sponged up miRNAs, similar to what miRNAs do for mRNAs [72]. Another study found that metastatic functions weaken when CDR1as regulates and works with the RNA binding protein known as IGF2BP3 [73]. Further research discovered CDR1 has the ability to prevent gliomagenesis by obstructing the p53/MDM2 complex [74]. Several studies have been published about the interaction between circRNAs and miRNAs, specifically Sry circRNA and CiRS-7 [71]. One study showed that Sry, a gene that decides the sex of mice, can be transcribed into circRNA [75]. Another study demonstrated that Sry circRNA behaved as a miRNA sponge and had 16 sites that can bind miR-138 [16].

### 3.3. Role in Diagnosis of Diseases: Cancer

Circular RNAs have been found to play important functions in cancer [115]. In cancers such as esophageal squamous cell carcinoma [97] and colorectal cancer [49], evidence consists of dysregulated circRNAs. Colorectal cancer was found to be the third highest cancers in both men and women compared to other cancers [116]. Like most other cancers, colorectal cancer is characterized by two factors: epigenetic as well as genetic modifications. Studies have shown that cancer progression or tumor formation have aberrant miRNA expression [117]. One study analyzed linear and circRNA expression and proliferation in tumor tissues. Researchers proposed that the tumor and typical colon mucosa samples from CRC patients had more than 1800 circRNAs. Tumor samples consistently had a lower circRNA to linear RNA isoforms ratio than colon samples that did not have the tumor. Samples from the colorectal cancer cell lines had an even lower ratio than the tumor samples. Idiopathic pulmonary fibrosis, proliferative diseases that are non-cancerous, and several other typical human tissues supported the correlation between proliferation and abundance of global circRNA [51].

The best methods for characterizing and identifying circRNAs are based on RIbo-Zero and RNase R treatment [118]. CircRNA being a new cancer biomarker needs more research, thus Vo et al. created MiOncoCirc as a resource for further research. More importantly, MiOncoCirc identified circ-*ACPP* and circ-*CPNE4* in prostate cancer. Studies have shown that circRNAs are also found in urine samples [42]. A variety of tumor tissues have displayed downregulation of circRNAs, supporting the possibility that several circRNAs have a role in suppressing tumors. However, it can also be interpreted as the dilution of accumulated circRNA when cell division occurs. Further analysis showed that cancer tissues produced more circRNA abundance compared to normal tissues due to several upregulated genes in cancer. Furthermore, researchers noticed a downregulation of circRNA in cells that proliferated across diverse types of tumors. This implies that a few circRNAs could have roles in suppressing tumors [51]. Additionally, determining the type of cancer could be possible if certain circRNAs that are upregulated in tissue can act as surrogate markers such as circ-*AURKA* in NEPC or circ-*AR* in CRPC [119].

### 3.4. Immune Response and CircRNAs

Studies have demonstrated that immune responses and diseases related to the immune system change expression of circRNA [120]. Regulating the immune cells’ function and differentiation is another role of circRNAs. Widespread circRNA expression was found in lymphoid differentiation, myeloid cells, and hematopoietic progenitors. Various enucleated blood cells aggregate additional circRNAs [121]. It has been found that dysregulation of circRNAs could be involved in the mechanisms that pathogens use to avoid immune response linked with relapse of amyloid leukemia (AML) and relapse blasts [122]. It was found that patients who had a relapse in acute myeloid leukemia (AML) had a specific circRNA profile compared to patients who were healthy or in complete remission. A recent study found that PKR is inhibited by cellular circRNAs. As previously mentioned, an innate immune response occurs when cellular circRNAs are degraded by endonuclease RNase L release PKR [120]. Several targeted DE-mRs signaling pathways found in the ceRNA network are involved in the evolvement of Wilms tumors (WT) because these pathways are linked with immune response and cell cycle [123].

### 3.5. Proteins and CircRNA

Here, we will discuss how circRNAs bind with protein, form circRNA–protein complexes, and interact with mRNA and sponge proteins. In recent years, accumulating evidence suggests that circRNAs, microRNA, and some proteins that bind with RNA play an essential role in gene function modulation that is connected to the development of various diseases [124]. CircRNAs modulate gene expression through their effect on RBPs and miRNAs [125]. Development of breast cancer in vivo and in vitro was found to be suppressed when introducing CREBZF mRNA nanoparticles. This contributes to new understandings of therapeutic approaches for breast cancer [126]. The mRNA translation and stability could be modulated by miRNAs binding to target mRNAs creating a complex with a protein known as Argonaute (AGO) protein [127]. Migration and proliferation of cells in prostate cancer were found to be caused by the interaction between RNA-binding protein FUS and circ0005276, which was formed through the backsplicing of the XIAP [128]. Another study found that circRNAs originating from exons in neuroepithelial stem cells (NES) in humans have a higher chance of having a genetic variation. Studies have shown that some circRNAs had increased in SFPQ ribonucleoprotein complexes and decreased expression due to SFPQ knockdown [129].

### 3.6. CircRNAs in Transcription and Splicing Regulation

As previously mentioned, circRNAs are capable of regulating the transcription of parent genes. It has been found that interaction between circURI1 and heterogenous nuclear ribonucleoprotein M (hnRNPM) caused alternative splicing genes to be regulated, leading to the inhibition of gastric cancer metastasis [130]. Another study has shown that EIciRNA-U1 snRNP and Pol II transcription complex interact with each other more at parental gene promoters, causing gene expression to increase [14]. The amount of cognate exon-6-skipped variant is raised when the host DNA locus and circSEP3 bind, thus creating an R-loop or RNA-DNA hybrid. This causes the recruitment of the splicing factor and pausing of the transcription [131]. A recent study has shown that circRNA: DNA associations have been found throughout the genome and are majorly present in MLL regions [132]. circMLL (9,10) is highly expressed in infants with leukemia and found to be associated with cluster regions of MLL breakpoints. circMLL (9,10) induces translocation and DNA breaks [132]. Interestingly, the circRNA circSMARCA5 was found to inhibit DNA damage repair by interacting with genes in the host. circSMARCA5 interacts with its host gene locus, which causes transcriptional pausing of its parent gene SMARC5 [133]. Additionally, circular intron RNAs (cirRNAs) can grow in human cells when they fail to debranch at the branchpoint of a 5′ splice site. This causes the expression of parental genes to increase by regulating the activity of elongation Pol II [15]. All this evidence suggests that circRNAs can regulate gene transcription and splicing of parental genes.

### 3.7. Role in Diagnosis of Diseases: Neurological Diseases

Researchers have found that circRNAs can be used as therapeutic targets or pathological biomarkers in various neurological diseases and manifestations [1]. A deteriorating nervous system can cause neuropathic pain [134]. It is one of the most common diabetic complications with a prevalence of about 30% of type II diabetic patients [135]. Both circRNAs and long-coding RNAs (lncRNAs) have demonstrated involvement in the neuropathic pain [136]. CircHIPK3, a type of circRNA and tumor suppressor, has been found to sponge numerous miRNAs, thus regulating the growth of cancer cells [105,108,109,137,138]. Targeting miR-124 to silence circHIPK3 diminished neuropathic pain in diabetic rate, indicating that a change to the circHIPK3/miR-124 axis contributes to the reduction of diabetic neuropathic pain progression [139]. An abundance of circHIPK3 was found in two places: (1) the root ganglion from diabetic rats that were STZ-induced and (2) serums from diabetic patients who go through neuropathic pain.

Dendrites and neuropils in the brain consist of enriched circRNAs that aid in controlling neural plasticity and synaptic function. This suggests that circRNAs could have a crucial role in diseases such as Alzheimer’s diseases (AD), Parkinson’s diseases (PD), and epilepsy that affect the nervous system [140]. In PD, a study revealed that α-synuclein expression is downregulated by miR-7. High expression of α-synuclein in the brain is greatly entangled in pathogenesis of PD. Additionally, cells are shielded against oxidative stress because miR-7 induces the downregulation of the α-synuclein protein that is induced [141]. MiRNA-7 specifically targets the nuclear factor (NF)-κΒ signaling pathway, thus sheltering against cell death that is caused by 1-methyl-4-phenylpridinium [142]. Furthermore, studies have provided evidence that miR-138 controls acyl protein thioesterase 1, which affects the brain’s ability to memorize and learn [143,144]. In brains with AD, miR-7 expression is upregulated by the functional deficiency of CDR1as and causes ubiquitin-protein ligase A, an AD-relevant target, to downregulate [145]. Ubiquitin-protein ligase is reduced in the AD brain and is critical for clearing amyloid peptides [146], thus suggesting that CDR1as may be important in the pathogenesis of AD [145].

### 3.8. Role in Diagnosis in Diseases: Autoimmune Diseases

The lack of immune tolerance to self-antigens and an impaired immune system are defining characteristics of autoimmune diseases. CircRNAs working as non-invasive biomarkers in diseases such as rheumatoid arthritis (RA), systemic lupus erythematosus (SLE), and multiple sclerosis (MS) have been demonstrated in recent studies [147]. It was demonstrated that patients with RA have upregulated expression of ciRS-7. Additionally, miR-7 roles were stopped by ciRS-7, causing reduced effects of miR_7 suppressing mammalian target of rapamycin (mTOR) [148]. Studies found levels of hsa_circ_0092285 and hsa_circ_0058794 increased while hsa_circ_0088088 and hsa_circ_0088088 levels diminished in peripheral blood mononuclear cells (PBMCs) of RA patients [149]. In MS patients who relapsed and remitted, it was found that MS pathogenesis was associated with aberrant RNA metabolism because PBMCs displayed dysregulated circRNA and alternative splicing isoforms [150]. The lncRNA known for controlling alternative splicing, more formally known as metastasis-associated lung adenocarcinoma transcript 1 (MALAT1), was upregulated in patients with MS [151,152]. Furthermore, thousands of alternative splicing events were found to be regulated. MALT1-knockdown Jurkat T cells also consisted of many differentially expressed circRNAs. Results from various studies suggested that MALAT1 dysregulation could result in the development of MS because it affects backsplicing (BSJ) and splicing events [153].

Approximately, 127 differentially expressed circRNAs were identified in patients with SLE [154]. It has been found that the three following circRNAs were expressed in SLE patients’ plasma: hsa_circ_100226, hsa_circ_102584, and hsa_circ_400011 [154]. Among the several miRNA response elements that were found in hsa_circ_100226, it was found that in chondrocytes, the suppression of p65 expression by miR-138 improved NF-κB activation. Osteoarthritis (OA) progression was found to be influenced by miR-138-5p, controlling inflammation and extracellular matrix catabolism [155]. Moreover, it was also found that CD4^+^ T cells in patients with SLE consisted of several downregulated and upregulated cricRNAs. DNA methyltransferase 1 expression improved, while CD70 and CD11 expression on CD4^+^ T cells in active and inactive SLE patients was reduced due to the downregulation of circRNA. Autoantibody production in SLE was stimulated because of CD70 and CD11a overexpression. [156]. In PBMCs, circlBTK is downregulated, while miR-29b is upregulated in SLE patients. CirclBTK binding together with miR-29b may prevent DNA demethylation and protein kinase from activating in SLE [157]. Much research has revealed that functions of immune cells can be controlled by protein kinase signaling pathway. Dysregulation of the pathway prompts SLE to advance quickly [158]. With a myriad of supporting evidence, it is thought that circRNAs have the capability to act as a non-invasive biomarker of SLE [159].

## 4. Circular RNA’s Role in Pathogenic Infection

### 4.1. CircRNAs in Viral Infections

circRNAs can be potential biomarkers in diagnosing different illnesses, but the expression of some host circRNAs is different after viral infections. In recent years, there has been tremendous development in approaches of bioinformatics and transcriptomic sequencing technology. This led to the advancements in recognition of various individual host circRNAs expressions in viral infections and viral circRNAs [160]. Methods used to assist in detecting differentially expressed circRNAs in tissues include two methods: sequencing and single-cell separation [161]. Additionally, by using developed methods, circRNA’s presence in fluid or tissue can be successfully detected. These methods can also be used to successfully determine the distribution and expression of both host and viral circRNAs.

Nasopharyngeal carcinoma (NPC) is challenging to diagnose because it is induced by Epstein–Barr virus (EBV) infection, and early detection of NCP could be lifesaving for patients affected by this type of cancer. CircRNAs such as hsa_circRNA_001387 (host) and circRPMS1 (EBV-encoded) can be specifically diagnosed in NPC tissues that have EBV infection compared to NPC tissues that are EBV-negative. This makes both hsa_circRNA_001387 (host) and circRPMS1 (EBV-encoded) diagnostic biomarkers of NPC [162]. Moreover, RNA-seq analysis of samples in chronic hepatitis B (CHB) displayed 99 circRNAs expressed differently [163]. Furthermore, results demonstrated various expression patterns of ~226 circRNAs in patients with hepatocellular carcinoma (HCC), which engage in the pathogenesis of the disease. Of the 226 circRNAs, downregulation of 37 circRNAs and upregulation of 189 circRNAs occurred in HCC patients’ samples [164]. Another study used RNA-seq data of HPV-infected and HPV-positive cancer cell lines to identify circE7. It has been suggested that circE7 could be a possible biomarker for HPV linked to high-risk cancers because of the favorable amount of E7 oncoprotein it can express [165]. The RNA-seq dataset further found circRNAs encoded in SARS-CoV-2, SARS-CoV-1, and MERS-CoV, which will be discussed more in-depth when looking at circRNAs in the coronavirus [4].

Studies have shown that circRNAs can be used as therapeutic agents. Certain features of circRNAs make them potentially useful in the development of vaccines. Vaccines are mostly mRNA, DNA, and cells based on different specific characteristics [166]. There are several characteristics that make circRNA-based vaccines more unique compared to typical vaccines. First, circRNA vaccines and mRNA vaccines are similar in terms of both being able to translate into protein in the cytoplasm. Therefore, it is unnecessary to integrate it into the genome making it safer. Second, half-lives in circRNAs are much longer than those of mRNA vaccines because of circRNA’s circular structure. Proteins can be translated from circular structures for extended dates. Up to now, only several endogenous circRNAs have demonstrated capabilities of promoting as a template for the protein translation [167]. Furthermore, dendritic cells can be stimulated by a tiny concentration of specific circRNAs [168,169,170]. Lastly, dendritic cells produced to express specific antigens with tremendous immunogenicity were presented by synthetic circRNAs [168,171]. Autoimmune responses have been challenged by circRNAs involvement; however, the specific synthesis and design of circRNA vaccines can be used to reduce immune responses that are undesirable [172,173]. In conclusion, circRNAs may play a critical part in viral infections as a therapeutic agent and vaccine.

As previously mentioned, some of the many roles of circRNAs include its behavior as a sponge miRNA and its effects on protein translations [171,174]. In viral infections, research demonstrates circRNA being used to modulate immune systems. CircRNAs can be a possible new antiviral treatment approach if they target and impair viral miRNAs. In HCC cells and tissues that are linked to HBV, circ_ATP5H and TNFAIP3 were found to be more commonly expressed. miR-138-59 expression was downregulated in these HBV-infected tissues and cells, suggesting that circ-ATP5H could be considered as a potential therapy option for the HCC associated with HBV because ATP5H functions as a miR-138-5p sponge and controls the expression of TNFAIP3 [175]. Another study found that circEZH2 targeting miR-22 can prevent mitochondria from opening a permeable pore that is targeted by gastroenteritis coronavirus. However, porcine intestinal epithelial cell damage and NF-kB activation were the results of transmissible gastroenteritis coronavirus downregulating circEZH2.

### 4.2. CircRNAs in Bacterial Infection

CircRNAs have displayed many roles in bacterial infection. It is speculated that one method of battling against bacterial infections is using EVs because bacterial infections continue to be one of many primary global health issues [176]. Several studies have determined that circRNA has a close relationship with bacterial infections [174,177,178]. Studies have shown the presence of circRNAs in milk-derived EVs (MEVs) for the first time. High-throughput sequencing demonstrated that *S. aureus* infection substantially changed the profiles of circRNAs in EVs. It also showed that these circRNAs were selectively packaged into EVs. Furthermore, differentially expressed circRNAs involvement in immune functions through bioinformatic analyses was proposed [179]. Therefore, hosts use EVs to transfer functional circRNAs in response to bacterial infection. Current research shows that circRNA plays a large part in many pathological processes, and its dysregulation in the progression of numerous diseases has caused the relationship between tuberculosis and circRNA to attain more attention over the years. *Mycobacterium tuberculosis* (Mtb) infection causes tuberculosis (TB) and continues to be a prevalent, deadly, and communicable disease [180]. PBMCs from patients with TB infection had more downregulated circ_0009128 and circ_0005836 as compared to patients who did not have TB infection, meaning that circRNA could potentially be a new biomarker for this active disease [181]. Through an extensive microarray analysis on the differences in expression patterns of circRNAs in PBMCs, evidence demonstrated that dysregulated circRNAs potentially used as biomarkers for interactions between TB and circRNA_101128-let-7a could have a large role in how PBMCs react to Mtb infection [182]. Additionally, by using whole transcriptome sequencing, it was revealed that patients with TB had many circRNAs that were differentially expressed in leukocytes [183]. Another study also proposed the idea of having numerous circRNAs act as biomarkers as a way of differentiating viral from non-viral pneumonia [174]. In conclusion, circRNAs are an important potential biomarker and tool to combat bacterial infection.

### 4.3. Cellular Circular RNA during Virus Infection

#### 4.3.1. CircRNA in Respiratory Syncytial Virus

Although induced modifications in cellular circRNA transcriptomes and viral circRNA expression have been found in DNA virus infections [184], those in respiratory syncytial virus (RSV) are not fully understood. RSV is a negative and single-stranded RNA virus that infects mainly children across the world and progressively becomes an important pathogen in the elderly [185]. RSV infection causes differential viral and cellular circRNA expression [184]. Yao et al. performed an expression analysis and identification on both viral and cellular circRNAs in A549 RSV-infected cells. Bioinformatic analysis identified over 2280 cellular circRNAs that were differentially expressed. In this analysis, three considerable expression clusters (UP3, UP1, and UP2) were found. Further results demonstrated that the best 10 circRNAs from the UP1 cluster were upregulated in another RNA virus infection poly (I.C) treatment. In turn, the same 10 circRNAs prevent RSV from replicating, and the same circRNA sequencing detected 1254 viral circRNAs. Both A549 cells and Hep-2 cells displayed RSV infection-induced viral circRNA expression. Results reveal viral or cellular circRNAs as potential new biomarkers or therapeutic targets, fostering a better understanding of host-RSV interactions.

#### 4.3.2. CircRNAs in Herpes Simplex Virus 1

A human pathogen known as herpes simplex virus type 1 (HSV-1) is part of the subfamily *Alphaherpesvirinae* and known for causing cold sores [186,187]. Virion host shut-off (*vhs*) protein is essential in HSV-1 lytic infections. It can quickly split viral and cellular circRNA when delivered to infected cells via incoming virus particles [188] and cleave circRNAs consisting of only an internal ribosome entry site (IRES) [189]. Recent research has found that linear mRNA degradation mediated by *vhs* prompts an enhancement of circRNAs compared to linear mRNAs during HSV-1 infection. Most circRNA enrichment is due to a decrease in linear RNA rather than an increase of circRNA synthesis. On the contrary, IAV and HSV-1 infection both demonstrated synthesis of circRNA arising from NEAT1_2, which is the long isoform of the nuclear paraspeckle assembly transcript 1 (NEAT1). This was linked with circRNA downstream and new linear splicing of NEAT1_2 induction within [186]. NEAT1_2 has critical paraspeckle structure because it creates a scaffold by being bound via numerous proteins involved in the process splicing, transcription, and RNA processing [190,191,192]. Friedl et al. demonstrated that upregulation and splicing of NEAT1_2 could be caused by ectopic co-expression of ICP22 and ICP27, which are both HSV-1 proteins, connecting to elevated NEAT_2 splicing and expression. Further analysis through published RNA-seq data found that poly(A) read-through analogous and NEAT1_2 splicing induction are analogous to IAV and HSV-1 infection in cancer cells upon knockdown of CDK7 or MED1 subunit of the Mediator complex that was phosphorylated by CDK7. In conclusion, research demonstrated new linear and circular NEAT1_2 splicing isoforms induction as typical components of IAV and HSV-1 infections and CDK7’s potential function in both infections [186].

#### 4.3.3. CircRNAs in Coxsackievirus Group B Infection

Coxsackievirus group B (CVB) is part of the *Picornaviridae* family that causes an array of diseases such as encephalitis, pneumonia, hepatitis, myocarditis, aseptic meningitis, and gastrointestinal illness [193]. Interaction between host cellular factors and viral components is a critical part in the pathogenesis of CVB [194]. Past research has found that modulation of CVB infection inflammatory response by miR-10a* and miR-146a respectively promotes CVB replication [195,196]. Up to now, whether circRNAs in CVB replication is involved or not remains unexplained. Hsa_circ_0000367 (circSIAE) is the only circRNA that has been disclosed to stop CVB3 (CVB type 3) replication by targeting cellular TAOK2 via sponge absorption of miR-331-3p [197].

Recent emerging evidence has shown that the pyroptosis process specifically seen in diabetic cardiomyopathy is regulated by circ_0076631 [198]. Qin et al. found that there was significant CVB type 3 (CVB3) replication when promoted by circ_0076631. Further analysis demonstrated indirect interaction of circ_0076631 with CVB3 is led by miR-214-3p sponging, which specifically aims at the CB3 genome 3D-coding region, so viral translation was repressed. Circ_0076631 knocked out induced CVB3 infection to be suppressed, making circRNA a possible target [199].

### 4.4. Viral Circular RNA (DNA/RNA Virus)

#### 4.4.1. CircRNA in Epstein–Barr Virus

EBV infection occurs in over 90% of the adult population globally. Currently, no vaccine is available to treat EBV infection. EBV infections have been associated with infectious mononucleosis, Hodgkin’s lymphoma, Burkitt’s lymphoma, and nasopharyngeal carcinoma. Studies have shown that miRNA, lncRNAs, and EBV-encoded small RNAs are non-coding RNAs (ncRNAs) that are produced in EBV during infection [57]. EBV circRNAs that were expressed across latency types imply that circRNAs play a role in intrinsic processes during latent infection. Evidence has demonstrated that an additional layer of the EBV transcriptome was identified through identifying latent and lytic viral circRNAs. Furthermore, EBV-encoded circRNAs expressed during latency are confined to type III latency. It could contribute to particular mechanisms such as Cp promoter regulation, promoting Cp-initiated transcript diversity, or other functions related to the type III latency [200].

#### 4.4.2. CircRNAs in Kaposi Sarcoma Virus (KSHV)

KSHV can cause many diseases, including a plasmablastic form of Castleman’s disease (MCD), primary effusion lymphoma (PEL), and Kaposi’s sarcoma. Like many other viruses, KSHV encodes miRNAs and aims for human or viral transcripts [201]. Several noncoding RNAs (ncRNA) are occasionally found during latency [202,203]. KSHV, along with EBV, is one of the few known human cancer viruses. It is also of one the few human viruses that express multiple viral miRNAs [204]. Studies have shown that polyadenylated nuclear (PAN) RNA was greatly expressed as early as lytic transcript. There were also reduced amounts during latency. Research demonstrated that PAN RNA locus created numerous RNase R-resistant circRNAs that were backspliced and low in abundance that stem in antisense and sense directions. This is coherent with new hyperbacksplicing processes [205].

#### 4.4.3. CircRNA in Human Papillomaviruses (HPV)

HPVs are circDNA viruses that are small and double-stranded. They infect epithelial cells in humans and are linked to numerous malignant and benign lesions of skin and mucosa [206]. Even though some evidence has shown that HPV infection has a critical part in the pathogenesis of about 5% of cancers in humans, the comprehension of latent infections to cancers risky HPVs advancement is still not understood [207]. Recent evidence has shown that oncogenic HPVs produce circRNAs, which includes circE7. Northern blotting and RT-PCR of cells transformed by HPV16 were both used to identify HPV16 circE7, whereas TCGA RNA-Seq datasets only identified HPV18 circE7. Furthermore, translated circE7 can form E7 oncoprotein, and E7 protein levels can be reduced due to disruptions of circE7 in CaSki cervical carcinoma cells, thus hindering cancer cells from growing in tumor xenografts and in vitro [153].

#### 4.4.4. CircRNA in Coronavirus

The continuing COVID-19 pandemic is caused by a single-stranded RNA virus known as severe acute respiratory syndrome coronavirus 2 (SARS-CoV-2). This positive sense RNA virus is part of the *Coronaviridae* (CoV) family [1]. Many studies have been performed on the transcriptomic analysis of SARS-CoV-2 infected cells to analyze the host and virus-encoded RNA. It is essential to analyze virus-encoded circRNAs and their correlation with the disease progression that will help the scientist to speculate the role of virus-encoded circRNAs in the disease manifestation and severity. A study found that in the late stages of coronavirus infection, bioinformatic analysis of virus RNA sequencing data identified 720 circRNAs in SARS-CoV-1, 28,754 circRNAs in MERS-CoV, and 3427 in SARS-CoV-2. Moreover, another RNA-seq study identified 224 circRNAs from SARS-CoV, 2764 circRNAs from MERS-CoV, and 351 circRNAs from SARS-CoV-2. SARS-CoV-2-encoded circRNAs differ from host-encoded circRNAs in several ways. A study found that circRNAs derived from host genome-derived are not as long and abundant as the circRNAs derived from the coronavirus [1]. Studies from Yang et al. and Cai et al. have provided excellent information using empirical evidence and bioinformatics analyses that a new group of circRNAs was produced from the genomes of coronavirus [1]. Additionally, 75 possible circRNAs derived from SARS-CoV-2 were detected from Vero E6 cells infected with SARS-CoV-2. Two circRNAs were a critical part of the coronavirus transcriptome because they had M and ORF3a containing powerful IRES signals, which was further demonstrated with sequence analysis [208].

Previously stated evidence using bioinformatics analyses demonstrated several opposing pieces of evidence about the expression, strand preference, and abundance of CoV circRNAs. It was associated with datasets, circRNA analysis pipeline, and strategy Cai et al. used [1]. A study has shown that circ_3205 was synthesized from the N gene part of SARS-CoV-2. Results revealed that circ_3205 causes SARS-CoV-2 to progress due to three reasons: (1) circ_3205 sponges hsa_miR_298, (2) circ_3205 helps cause the upregulation of PRKCE and KCNMB4 mRNAs, and (3) circ_3205 causes the progression of SARS-CoV-2 infection because circ_3205 acts as a ceRNA [209]. Another study has shown that an upregulated human miRNA in SARS-CoV-2 is ORF3a hsa-miR-5896-5p. ORF3a hsa-miR-5896-5p upregulation is possibly the most important cause of ORF3a expression reduction during infection because ORF3a is linked with apoptosis. Apoptosis and ORF3a connection is crucial to the host antiviral defense’s ability to control viral infection [210]. A ceRNA network was proposed to be embroiled with Ddx68, Ppp1r10 and C330019G07RiK, Gm 26917, and miR-124-3p, thus demonstrating miRNA-circRNA-mRNA networks’ importance in gene expression when there are new possible target genes in SARS-CoV-2 infection [211].

## 5. Characteristics of Viral Circular RNA

Human cytomegalovirus (HCMV) replicates in the epithelial cells of the organ that it first infects and spreads through the blood to other organs. HCMV infections are the leading cause of birth defects. In the immunocompromised population, complications that are life-threatening and congenital anomalies can be caused if latent HCMV reactivates [212]. A recent study stated that the newest group of HCMV transcripts are circRNAs. Bioinformatics predicted that 704 circRNAs are encoded by the TB40/E strain while 203 circRNAs are encoded by the HAN strain. The researchers also systematically compared the following circRNA characteristics: (1) breakpoint motif, (2) length distribution, (3) the number of exons between viral circRNAs and circRNAs encoded by the host genome, and (4) strand preference [208].

### 5.1. Breakpoint Motif of HCMV CircRNAs

AG|GU was stated as the conservative splicing signatures of acceptor sequences and splicing donor that human circRNAs consisted of [213]. Because circRNA’s distinct splicing signatures that are transcribed from various types of genomes are obscure, CIRI2, an impartial algorithm, was used to evaluate data sets that already exist to determine HCMV-encoded circRNAs. CircRNAs from KSHV and EBV were also evaluated by CIRI2 [208,214]. The evidence indicates that HCMV circRNAs consist of backsplicing junctions that have a consensus motif. This is analogous to what was identified in EBV and KSHV.

### 5.2. Length Distribution, Exon Numbers, and Strand Preference

Length, exon numbers, and strand preferences may vary in circRNAs between host and viruses. A recent study performed an in-depth analysis of viral and host circular RNAs to delineate these characteristics of circRNAs in hosts and viruses. The nucleotide length of circRNAs found in chickens, rabbits, mice, macaques, rats, and humans ranges from 250 to 500 nucleotides. CIRI-full, a de novo reconstruction of the entire length of circRNAs, was used to further define circRNAs in EBV, KSHV, and HCMV [214,215]. Varying numbers of full-length circRNAs were identified in KSHV (283 circRNAs), EBV (64 circRNAs), and HCMV (145 circRNAs). Extra partially assembled viral circRNAs were also found [208]. A study has reported that the lengths of HCMV circRNAs were analogous to other viral circRNAs in DNA viruses. The standard lengths of circRNA in KSHV (377.7 nt), HCMV (372.0 nt), and EBV (454.2 nt) were longer than the length of the host [208]. Yang et al. further examined the characteristics of circRNA in HCMV by performing bioinformatics statistical analysis on circRNA’s strand preferences. Results showed that in humans, HCMV circRNAs usually derive from DNA that are positive-stranded instead of circRNAs [208]. In B lymphoma cells, KSHV and EBV enact latency [200,216]; however, positive-strand DNA creates a majority of the circRNAs in EBV. Negative-strand DNA creates the most circRNAs in KSHV. This indicates that there is genomic strand preference in viral circRNAs [208].

To verify the number of exons enclosed in EBV, KSHV, HCMV, and host cells, Yang et al. used a statistical analysis of the circRNA exon numbers that were at full length and reconstructed. circRNAs that included a single exon made up about 80% of the circRNAs in KSHV and HCMV circRNAs. Multiple circular exons in circRNAs made up about 86% of EBVs. Host circRNAs also contain several circular exons [208].

## 6. Functions of Cellular/Viral Circular RNA

CircRNA’s fascinating biology allows it to have numerous functions in the cell and a critical role in numerous diseases ranging from cancer to pathogenic infections to DNA and RNA viruses. Notable functions of circRNA include acting as a miRNA and its role in cell proliferation. Here, we will take a deeper dive into the specific functions of cellular and viral circRNAs.

### 6.1. Biological Effects Caused by Protein Binding

When circRNAs bind to specific proteins in the cytoplasm, it may incite the induction of extrinsic stimuli and alter RNA translation, transportation, maturation, and transcription [160]. Transcription factors, proteases, and RNA processing proteins are common RNA-binding proteins. CircRNAs can bind to these RNA-binding proteins (RBPs) and assist in controlling protein translation and expression of genes [217,218]. Table 2 lists the circRNA that binds with proteins and affects their activity. There is substantial evidence that supports the interaction between RBPs and circRNAs even though there are fewer binding sties for circRNA-binding proteins than mRNAs binding proteins [219]. NF90/NF110 promote circRNA synthesis because they have the ability to stabilize the intronic complementary sequences when bound to RNAs. The structure of these immune factor proteins consists of a double-stranded RNA binding site. Both proteins create in the cytoplasm a circRNP complex and cooperate with endogenous circRNAs. Studies have shown that when antiviral proteins like NF90/NF110 bind to circRNAs under normal circumstances, unwanted innate immune responses may be suppressed. In contrast, NF90/NF110’s inclination to bind to circRNAs causes these proteins to be stored for quick immune responses that fight off viruses [44]. When viral infection occurs, NF110 and NF90 bind to viral mRNA after it discharges the circRNP complex. This prohibits viral translation in infected cells [120].

### 6.2. Platforms/Sponges for Proteins

In neuronal tissue, evidence demonstrates that intronic sequences drive how quickly animal circRNAs are produced. Furthermore, canonical pre-mRNA splicing, and circularization of exons contend against each other, which are kept in animals and specific to the tissue. Muscle blind binding sites are only bound by MBL and are contained in both introns flanking circMbl and circMbl itself. MBL binding sites influence how circMbl biosynthesis is affected by how MBL levels modulate [40]. Similarly, the nucleus of long-term hematopoietic stem cells (LT-HSCs) from the bone marrow consists of highly expressed cis-cGAS [229]. cGAS, located in both the cytoplasm and nucleus, identifies a large array of self-originated and pathogenic DNAs [230,231]. Evidence demonstrated that when there is a deficiency of cia-cGAS in mice, it leads to reduced dormant LT-HSCs levels and increased type 1 IFN expression in bone marrow. In the nucleus, cia-cGAS synthase activity was blocked when it attached to DNA sensor cGAS when it was under a homeostatic environment. This process assured that inactive LT-HSCs would not experience exhaustion mediated by cGAS. Furthermore, the creation of type 1 IFNs found in LT-HSCs mediated by cGAS was inhibited because the binding affinity cia-cGAS has for cGAS is stronger than the binding affinity self-DNA has to cGAS [229].

### 6.3. CircRNA Expression Abundance

CircRNAs have the ability to directly bind to mRNA to affect the abundance of gene expression. *circZNF609*, also known as circRNA zinc finger protein 609, is translated to a protein to regulate myoblast proliferation [167,232]. When expressed circZNF609 interacts with mRNA, it prefers recruiting ELAVL1, causing circZNF609 to enhance translation and stability. This can be seen with *CKAP5* mRNA interaction. The following several processes occurred: (1) the role of microtubule in cancer cells was controlled; (2) translation of CKAP5 was enhanced; (3) progression of cell-cycle was sustained; and (4) backsplicing junction overlapped [233]. However, it is still uncertain how the *circZNF609-CKAP* mRNA is regulated with other functions of circZNF609, including its role in translation [167], functioning as a miRNA sponge [234] or with the *circZNF609* m^6^A modification that specifically selects methyltransferase-like 3, and promoting black-splicing with YTHDC1 [56]. Furthermore, circRNAs affect maternal gene expression by working with U1 snRNP and controlling RNAPol II elongation activity/TER [235]. RNA Pol II is linked to both circular poly(A) binding proteins and EIF3J in human cells [14].

### 6.4. Competing with Linear mRNAs

mRNA translation can be altered when circRNAs outperform mRNAs by interacting with proteins that are similar [220,236,237]. Studies have found that competition between circularization and splicing can occur when circRNAs synthesize during RNA processing. For example, evidence revealed that exon2 from the MBL gene circularizes circ_Mbl. Circ_Mbl synthesis is induced when the circ_Mbl gene and the sequences of pre-mRNA and circ_Mbl bind together. However, circ_Mbl-MBL binding competitively suppresses canonical mRNA splicing [40]. In another study, there are about 12 expressed circPABPN1 per cell that prevents the binding of HuR to PABPN1 and mRNA. It was demonstrated through polysome analyses that *PABPN1* translation is positively maintained by HuR, therefore suggesting that the binding of *circPABPN1* and HuR can suppress HuR-promoted *PABPN1* mRNA translation [220]. Furthermore, Exons 2 to 5 from pre-mRNA *FAM120A* created *circFAM120A*. *CircFAM120A*s are oncogenic RNA, mostly located in mono ribosomes that stimulate cell proliferation. They inhibit *FAM120A* mRNA from attaching to IGF2BP2 and the m^6^A reader. This causes FAM120A to efficiently translate, which is needed for the proliferation [236]. Most competitive binding that occurs between *FAM120A* mRNA and *circRAM120A* to IGF2BP2 takes place in monoribosomes. Approximately one-third to one-fifth of the total *FAM120A* mRNA is the total amount of *circFAM102*. The binding affinity of *circFAM120A* to IGF2BP2 is heightened because of the m^6^A on *circFAM120A* [236].

## 7. Conclusions

In this review, we briefly described circRNA metabolism, specifically detailing the localization, regulation, biogenesis, degradation, and modification of circRNAs. Circular RNAs have sequences overlapped with their cognate linear RNA, which makes identification and quantification difficult. However, new biological methods have been developed to detect and study circular RNAs. Circular RNAs have been detected in cytoplasm and nucleus, thus playing important roles in regulating genetic and cellular processes. Although an animal model is lacking to evaluate the role of circular RNA in various diseases and pathological conditions, it is crucial to study the correlation between disease progression and the abundance of circRNAs. Identifying circRNA’s role under different pathological and physiological conditions can influence the development of new therapeutics for disease treatment. We summarize the mechanisms of (1) how circRNAs function as a miRNA, (2) how circRNAs function in cell proliferation, and (3) cirRNA’s roles in cancer, neurological diseases, autoimmune diseases (SLE, RA, and MS), and pathogenic infection (bacterial and viral infections). We further delve into circRNA’s role in viral infections by discussing cellular circRNA during viral infection, viral circRNA in DNA/RNA viruses, characteristics of viral circRNA, and functions of cellular/viral circRNA. Recent studies have identified circRNAs in HCMV, SARS-CoV2, and other viruses, implying that circRNAs may function in virus replication, growth, and disease manifestation. It is imperative to identify circRNAs in other viruses and study their roles in disease progression. The presence of circRNAs in viruses created new opportunities for researchers to evaluate their biological functions and relevance to disease. Taken altogether, it is clear that circRNAs are critical in the pathological and physiological functions in human diseases and development due to their unique characteristics and functions as cellular and viral circRNA, which specifically include: (1) how protein binding produces biological effects, (2) how circRNAs act as protein sponges, (3) how binding to mRNAs affect circRNA abundance, and (4) how circRNAs compare to linear RNAs. In the future, we anticipate more roles of circRNA being discovered and intriguing circRNA-based applications being implemented in biomedical research and therapeutic approaches.

## Figures and Tables

**Figure 1 ncrna-09-00038-f001:**
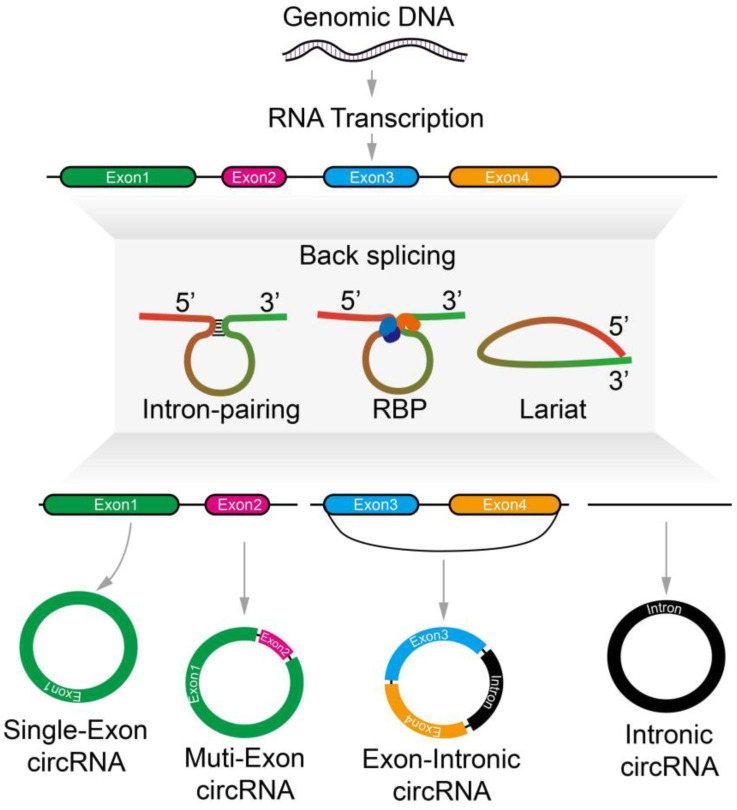
Biogenesis of different types of circRNAs. Transcription of genes produces pre-mRNA molecules. Pre-mRNA undergoes a splicing process to generate circular RNAs and mature mRNA. The circularization process is driven by different factors, including intron-paring, RBP, lariat, and others. Different types of circRNAs, such as single-exon (circRNA that contains a single exon), multi-exon (circRNA that contains multiples exons), exonic-intronic (circRNA that contains exons and introns), and intronic circRNAs (circRNA composed of introns), can be formed.

**Figure 2 ncrna-09-00038-f002:**
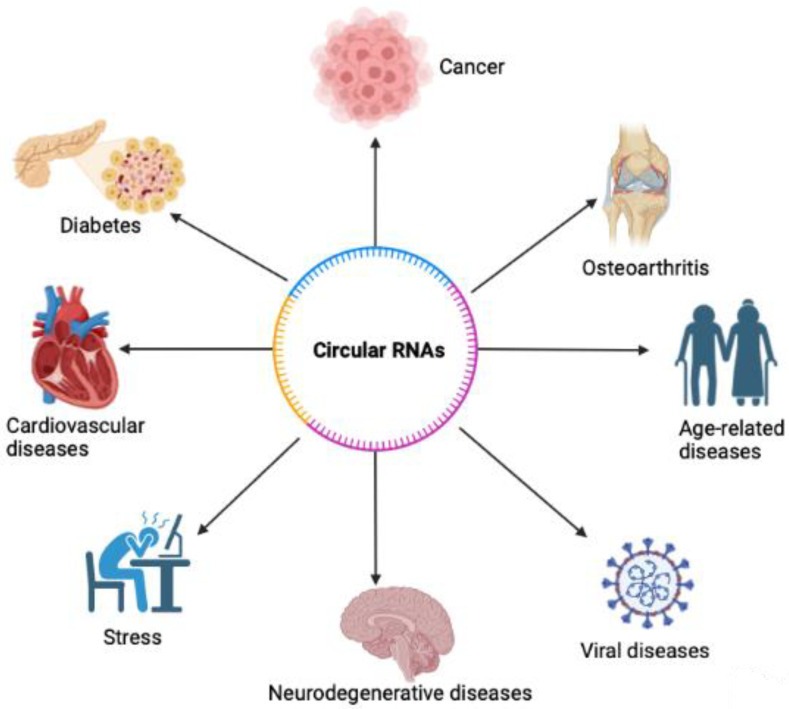
Circular RNAs have been associated with various diseases. CircRNAs have been found to play important roles in diseases, including diabetes, cardiovascular diseases, age-related diseases, osteoarthritis, cancer, stress, viral diseases, and others.

**Figure 3 ncrna-09-00038-f003:**
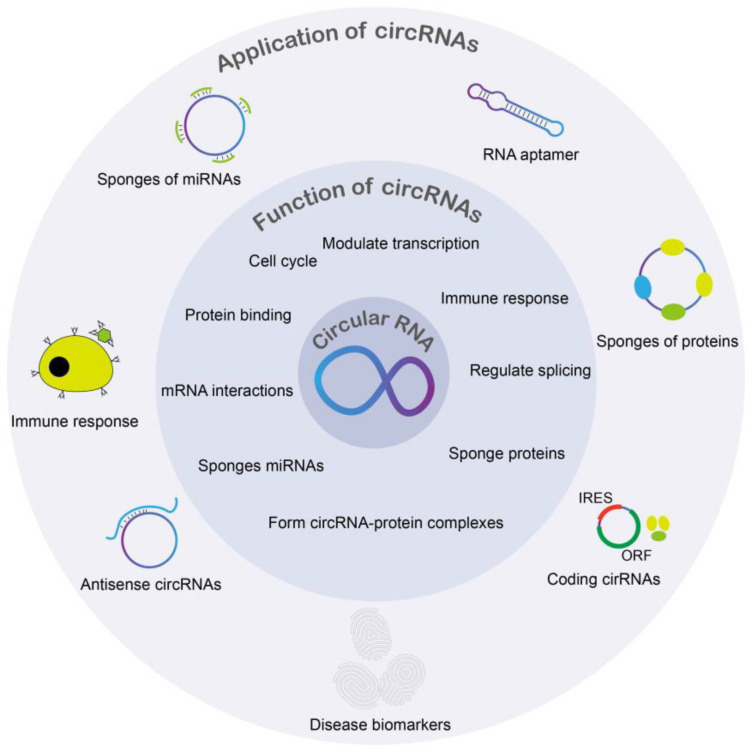
An overview of functions and biomedical importance of circular RNAs in different cellular processes.

**Table 1 ncrna-09-00038-t001:** List of circRNAs functioning as miRNA sponges.

Circular RNA	Functions	Targets	Interactions with Protein	Cell Type	Ref.
circRNA_0084043	Stimulates cancer progression	SNAIL	miR-153-3p	melanoma	[76]
circIRAK3	Promotes migration/invasion	FOXC1	miR-3607	Breast cancer cells	[77]
circBIRC6	Maintains pluripotency	SOX2, NANOG, OCT4	miR-145, -34a	hESCs, iPSCs	[78]
circATP2B1	Stimulates invasion	FN1	miR-204-3p	CCRCC	[79]
circLARP4	Prevents proliferation/invasion	LATS1	miR-424-5p	Gastric cancer	[80]
circADAT1 (circRNA_008913)	Decreases carcinogenesis	DAB2IP	miR-889	HaCaT	[81]
circACTA2	VSMC contraction	SMA	miR-548f-5p	HASMC	[82]
circPVT1	Stimulates proliferation	Aurka, mKi67, Bub1	miR-497-5p	HNSCC	[83]
Hsa_circ_0000799 (circBPTF)	Stimulates cancer progression	RAB27A	miR-31-5p	Bladder cancer	[84]
circCCDC66	Stimulates cancer progression	MYC, EZH2, DNMT3B	miR-93, -185, -33b	CRC	[85]
circNASP	Stimulates cancer progression	FOXF1	miR-1253	Osteosarcoma	[86]
Hsa_circ_0002052	Prevents cancer progression	APC2	miR-1205	Osteosarcoma	[87]
circTCF25	Stimulates cancer progression	CDK6	miR-107, -103a-30	Bladder carcinoma	[88]
circZFR	Stimulates cancer progression; prevents cancer progression	a) PTEN b) ZNF121 c) C8orf4 d) CTNNB1	a) miR-107, -130a b) miR-4302 c) miR-1261 d) miR-3619-5p	a) Gastric Cancer b) PTC c) Lung cancer d) HCC	[89,90,91,92]
circMYO9A (circRNA_000203)	Stimulates fibrosis	Col3a1, Col1a2, CTGF, SMA	miR-26b-5p	Cardiac fibroblast	[93]
circMTO1	Prevents cancer progression	HCC	miR-9	P21	[94]
circFBLIM1	Stimulates cancer progression	FBLIM1	miR-346	HCC	[95]
CDR1as	Myocardial infarction; Neural development; anti-oncogenic; stimulates proliferation/metastasis; osteoblastic differentiation insulin secretion;	a) MAGE-A b) HOXB13 c) EGFR CCNE1, PIK3CD, d) GDF5 e) P21 f) PARP, SP1 g) Pax6, Myrip h) Fox	a) miR-876-5p b) miR-7 c) miR-7 d) miR-7 e) miR-135a, -7 f) miR-7 g) miR-7 h) miR-67, -7	a) ESCC b) Islet cells c) NSCLC d) PDLSC e) Bladder cancer f) Cardiomyocytes g) ESCC h) Neural Tissue	[64,96,97,98,99,100,101,102,103,104]
circNT5E	Stimulates cancer progression	PIK3CA, NT5E	miR-422a	Glioblastoma	[65]
circHIPK	Stimulates proliferation/ migration; prevents cancer from progressing; β-cell function	a) AQP3 b) HPSE c) CDK6, ROCK1, d) FZD4, VEGF-C WNT2, e) Mtpn Slc2a2, Akt1, f) FAK, EGFR, IGF1R g) IL6R, DLX2	a) miR-124 b) miR-558 c) miR-124 d) miR-30a-30 e) miR-124-3p, -338-3p f) miR-7 g) miR-193a, -584, -29b, -654-193a, -124, -379, -152, -338, 29a	Cancer tissues	[63,66,105,106,107,108,109]
circWDR77	Stimulates proliferation	FGF2	miR-124	VSMC	[67]
circC1orf116 (circRNA_8924)	Stimulates cancer progression	CBX8	miR-519-5p, -518d-5p	Cervica tumor cells	[110]
circITGA7	Prevents proliferation/metastasis	NF1	miR-370-3p	CRC	[111]
circZNF609	Myoblast differentiation Retinal vascular dysfunction; neurodegeneration	a) MEF2A b) METRN c) BCLAF1	a) miR-615 b) miR-194-5p c) miR-615	a) Vascular endothelial b) RGC c) C2C12	[112,113,114]
SRY	Determines sex		miR-138	Testis	[16]

**Table 2 ncrna-09-00038-t002:** List of circular RNAs that act as a protein decoy.

Circular RNA	Cell Type	Interactions with Protein	Functions	Ref.
circPABPN1	HeLa	HuR	Inhibits PABPN1 translation	[220]
circMTO1	Breast Cancer Cells	TRAF4	Suppresses proliferation	[221]
circFOXO3	Heart Tissues, Non-Cancer Cells	P53, P21, ID-1, FAK, MDM2, HIF1α, CDK2, E2F1	Induces apoptosis Inhibits cell cycle progression Cardiac senescence	[110,222,223]
circAMOTL1	Cardiomyocytes	AKT1, PDK1	Supports cell survival	[224]
circSMARCA5	Glioblastoma	SRSF1	Suppressor of tumor	[225]
circDNMT1	Breast Cancer Cells	Auf1, P53	Promotes proliferation	[226]
circANRIL	Vascular Tissues	PES1	rRNA maturation	[227]
circHECTD1	Macrophage	ZC3H12A	Activation of macrophage	[228]

## Data Availability

Not applicable.

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
