# Peer review of "Functions of Circular RNA in Human Diseases and Illnesses"

_ncrna, 2023, doi:10.3390/ncrna9040038_

Round 1

Reviewer 1 Report

The review about the functions of circRNA in various diseases describe recent studies about the involvment of circRNA in the diagnosis and prognosis of different diseases and virus infections. 

The review is well written and accompanied by numerous examples of circRNA. Additionally, it has a big focus and detailed explanation of the circRNA functions during the development of different virus infections. 

I have several suggestions, which may improve this review:

1. Fig. 1 describes a lot of circRNA fucntions, but in the text only two types of them were desctibed. Please, add additional info about the circRNA, which are the examples of the other functions listed on Fig. 1.

2. Please, rename 3.2 section. The athors wrote that circRNA are the sponges of miRNA, so it's not the same function.

3. Please, add a small discussion about the similarity and differences between circRNA and lncRNA. Because both of them have similar function in the cell. 

Reviewer 2 Report

Functions of Circular RNA in Various Diseases and Illnesses

Circular RNAs (circRNAs)  have been found to affect many disease-relevant pathways through a diverse array of mechanisms, including forming R-loops, sponging proteins or miRNAs, and translating functional proteins, leading to different pathological phenotypes.

This work is a significant contribution to the field because it summarizes the essential functions in many different DNA viruses including Cytomegalovirus, Coronavirus, Epstein-Barr Viruses and others. Article is well organized and comprehensively described but more images are needed.

The studies carried out in 2022-2023 are missing from the bibliography (For example: Ikeda Y, Morikawa S, Nakashima M, Yoshikawa S, Taniguchi K, Sawamura H, Suga N, Tsuji A, Matsuda S. CircRNAs and RNA-Binding Proteins Involved in the Pathogenesis of Cancers or Central Nervous System Disorders. Noncoding RNA)
